# A novel scoring system to predict the requirement for surgical intervention in victims of motor vehicle crashes: Development and validation using independent cohorts

**Ryo Yamamoto** [ORCID] *, **Tomohiro Kurihara, Junichi Sasaki**

Department of Emergency and Critical Care Medicine, Keio University School of Medicine, Shinjuku, Tokyo, Japan

* ryo.yamamoto@gmail.com

## Abstract

### Background

Given that there are still considerable number of facilities which lack surgical specialists round the clock across the world, the ability to estimate the requirement for emergency surgery in victims of motor vehicle crashes (MVCs) can ensure appropriate resource allocation. In this study, a surgical intervention in victims of MVC (SIM) score was developed and validated, using independent patient cohorts.

### Methods

We retrospectively identified MVC victims in a nationwide trauma registry (2004–2016). Adults $\geq$ 15 years who presented with palpable pulse were included. Patients with missing data on the type/date of surgery were excluded. Patient were allocated to development or validation cohorts based on the date of injury. After missing values were imputed, predictors of the need for emergency thoracotomy and/or laparotomy were identified with multivariate logistic regression, and scores were then assigned using odds ratios. The SIM score was validated with area under the receiver operating characteristic curve (AUROC) and calibration plots of SIM score-derived probability and observed rates of emergency surgery.

### Results

We assigned 13,328 and 12,348 patients to the development and validation cohorts, respectively. Age, motor vehicle collision and vital signs on hospital arrival were identified as independent predictors for emergency thoracotomy and/or laparotomy, and SIM score was developed as 0–9 scales. The score has a good discriminatory power (AUROC = 0.79; 95% confidence interval = 0.77–0.81), and both estimated and observed rates of emergency

**Data Availability Statement:** The data related to this study were used under licence from the Japanese Association for Trauma Surgery and the

Japanese Association for Acute Medicine, and restrictions apply to the public availability of these data. Data are, however, available from the Japanese Trauma Care and Research (contact via email: jtcr-info@jtcr-jatec.org) for researchers who meet the criteria for access to confidential data.

**Funding:** The authors received no specific funding for this work.

**Competing interests:** The authors have declared that no competing interests exist.

surgery increased stepwise from 1% at a score $\leq 1$ to almost 40% at a score $\geq 8$ with linear calibration plots.

## Conclusions

The SIM score was developed and validated to accurately estimate the need for emergent thoracotomy and/or laparotomy in MVC victims.

## Introduction

Motor vehicle crash (MVC) is a major cause of mortality and morbidity across the world [1–3]. Despite the advent of sophisticated engineering technology and improvements in traffic infrastructure that aimed to reduce the incidence of MVCs, nearly 100 deaths and more than 6500 injuries are reported each day even in economically developed nations [4].

A retrospective analysis of mortality patterns associated with vehicular injuries using national data in the UK and the US revealed a significantly lower in-hospital mortality or morbidity in patients with severe injuries who received care at high-level trauma centres [5,6]; however, a population-based ecologic study found that facilities lacking surgical specialists including general- and neuro-surgeons round the clock showed increased MVC-related mortality [2]. Several retrospective studies regarding quality improvement also reported that delayed laparotomy for blunt abdominal trauma was associated with decreased survival [7,8], and some investigators have suggested timely thoracotomy to obtain favorable clinical outcomes [9,10].

Considering that timely surgical interventions improve outcomes for trauma victims following MVCs, the strategy for trauma care should include early hemostasis, decontamination of peritoneal infection, and early control of digestive fluid leakage, particularly outside the well-designed trauma system or designated trauma centres. Although several triage algorithms using mechanism of injury (MOI) and/or anatomic and physiologic criteria successfully identified patients who needed treatment at high-level trauma centres [11–13], they have not been validated to predict the need for emergency surgical intervention, particularly for laparotomy and thoracotomy. It should be emphasised that various scoring scales, such as Emergency Surgical Score (ESS) and Emergency Surgery Acuity Score (ESAS), have been developed to identify perioperative mortality or morbidity, but not to predict emergency surgical intervention [14,15].

Accordingly, we aimed to establish a universally acceptable grading system that can predict the requirement for emergency surgery, including laparotomy and thoracotomy, within 24 hours after MVCs. Thus, using a nationwide database, we developed a novel index, the surgical intervention in victims of MVC (SIM) score, which was examined to possess sufficient applicability in an independent cohort derived from the database. In this study, we hypothesised that the SIM score would possess sufficient calibration and discriminatory power to predict the need for surgical intervention in trauma victims following MVCs.

## Methods

### Study design and setting

This was a retrospective cohort study using data from the Japan Trauma Data Bank (JTDB), that was established as a Japanese nationwide trauma registry in 2003 and is maintained by the

Japanese Association for the Surgery of Trauma and the Japanese Association for Acute Medicine, with more than 200 participating major hospitals, including tertiary care centres. Data were collected prospectively and entered into the online data collection portal by treating physicians or volunteer registrars designated by each hospital. All collaborating hospitals obtained individual local institutional review board approval for the Conduct of Human Research before initiating the study (the Keio University School of Medicine, Ethics Committee approved the current study. Application number is 20090087). Requirement for informed consent was waived because of the anonymity of the data being used.

### Study population

Patients retrospectively identified in the JTDB were MVC victims who arrived at each participating centre between 2004 and 2016. Patients aged $\geq$ 15 years and with measurable systolic blood pressure upon arrival were included and those with missing data on the type of surgery, or unknown and missing dates of injury and/or surgery were excluded.

### Data collection

Data collected included age, sex, MOI, prehospital vital signs, vital signs upon arrival, any surgical procedures or angiography, date of injury, date of surgery, transfusion within 24 hours after arrival, Abbreviated Injury Scale score, Injury Severity Score (ISS), length of hospital stay and survival status during discharge. Emergency surgical intervention was defined as thoracotomy and/or laparotomy performed within 24 hours after arrival. Craniotomy and vascular surgery in extremities were not included as the emergency surgical intervention in this study because some anatomic and physiologic criteria, such as Glasgow Coma Scale (GCS) and massive bleeding and/or distal circulatory deficit in extremities have been reported to be able to predict the need for such surgical interventions [16,17]. Emergency hemostatic surgery was defined as surgical hemostasis performed within 24 hours after arrival. Emergency hemostatic interventional radiology (IVR) was defined as arterial embolization within 24 hours after arrival, which was performed under IVR prior to any surgical intervention. Additional hemostatic IVR was defined as arterial embolization performed within 48 hours after initial hemostatic procedure. Conflicting and/or ambiguous data elements were coded as missing data.

### Outcome measures

The primary outcome was the requirement of emergency surgery. Secondary outcomes included emergency hemostatic surgery, emergency hemostatic IVR, survival to discharge, and hospital-free days, also known as days alive and out of hospital, till day 90 after injury (combination of in-hospital death and length of hospital stay defined as the number of days alive and out of the hospital). Patients who died during the index hospitalisation and those who were hospitalised for >90 days were classified as having 0 hospital-free days. For patients who were discharged alive before day 90, hospital-free days were calculated as 90 minus length of hospital stay.

### Sample size estimation and data preparation

After patients were selected based on inclusion and exclusion criteria, they were assigned to the development or validation cohorts based on the date of injury; patients who were injured between 2004 and 2012 were in the model development cohort while remaining patients who were injured between 2013 and 2016 were allocated to the model validation cohort. Subsequently, development and validation of the grading index was performed following standard

methods of sample size estimation for multivariate logistic regression with at least 10 outcomes for each potential predictor analysed in the model. Thus, for an expected rate of emergency surgery of 3%–5%, at least 7000 patients (>200 emergency surgeries) were required in the development cohort to appropriately perform multivariate logistic regression with 20 potential predictors [18].

## Development of the SIM score

The scoring system was developed using multivariate logistic regression models. The primary outcome was entered in the model as a dependent variable and potential predictive variables were selected from known survival predictors in trauma victims and other clinical variables relevant to MVC [11–13,19–22]. Owing to clinical plausibility and practicality, age and vital signs were transformed to binary variables as follows: age (< 80 or $\geq$80 years) [23], GCS($\leq$13 or $\geq$14), systolic blood pressure (sBP; <110 or $\geq$110 mmHg) [22], heart rate (HR; <100 or $\geq$100 bpm), respiratory rate (RR; <22 or $\geq$22 /min).

The results from logistic regression modelling were used to identify predictors of the requirement for surgical intervention from the potential predictor variables assessed in the models. Weighted averages were used to assign scores to each selected predictor or variable in the final severity grading index termed the SIM score. The coefficients, namely, the odds ratio (OR) for each selected predictor variable were divided by the lowest common denominator and rounded off to the nearest half integer or nearest integer in several iterations to develop a simple score. A receiver operating characteristic (ROC) curve was constructed for each iteration to ensure consistency in the area under the ROC curve, also known as the c-statistic [15,24].

## Validation of the SIM score

The SIM score was validated using data from the independent validation cohort. Multivariate logistic regression analysis was used to assess discrimination power and calibration, wherein the requirement for emergency thoracotomy and/or laparotomy was designated as the dependent variable and the predictor variables of the SIM scale comprised the independent variables. Discrimination power was measured based on the c-statistic, and calibration was evaluated by the Hosmer–Lemeshow goodness-of-fit test [15,24].

Next, the practicality of the SIM score to enable prediction of the need for surgical procedure in MVC patients was examined by comparing the SIM score-estimated probability of emergency surgery with the observed rate of thoracotomy and/or laparotomy performed within 24 hours after injury [15,24].

To assess if the validation was independent of the method of comparison between the expected and the observed rates, sensitivity analyses were performed to confirm the robustness of SIM score. The SIM scores were analysed by logistic regression and linear regression, along with ISS and transfusion within 24 hours after arrival, for primary and secondary outcomes.

## Statistical analysis

Descriptive data are presented as mean ± SD, median (interquartile range), or number (%). Results were compared using unpaired t-tests, Mann–Whitney U tests, Chi-square tests, or Fisher's exact tests, as appropriate. All hypothesis testing considered a two-sided α threshold of 0.05 as statistically significant.

In the development cohorts, missing values were replaced with estimated values using a multiple imputation method, in which five sets of plausible values were calculated by linear regressions and then aggregated into estimated values. Variables that have considerable

numbers of missing data (>10% of all cases) were not used as potential predictors to maintain the reliability of analyses [25,26]. In multivariate logistic regression analyses, variables were entered using the simultaneous method. Interactions between important predictor variables in the final models were analysed by including interaction terms in logistic regression models. Statistically significant interactions were determined with two-sided $\alpha < 0.01$, while nonsignificant interactions were not subsequently used. Variance inflation factor (VIF) was used to assess multicollinearity, wherein VIF greater than 3 was considered statistically significant [27]. Any correlations between variables that were not used due to high number of missing data and predictors selected by the logistic model were examined using Pearson or Spearman correlation methods as appropriate to enhance the generalizability of the SIM score.

All statistical analyses were conducted using the SPSS version 24 (IBM Corp., Armonk, NY) and Microsoft Excel (Microsoft, Redmond, WA).

## Results

### Patient characteristics

The patient flow diagram is summarised in Fig 1. Database screening identified 31,936 patients who had suffered from MVCs and presented to collaborating hospitals during the study period. Among them, 29,977 patients were aged ≥15 years, and 30,083 patients presented with measurable blood pressure at all participating centres. A total of 28,124 patients satisfied all the inclusion criteria, among whom 2,410 were excluded due to missing data on the type of surgery or the date of surgery, and 38 were excluded due to unknown date of injury; thus, 13,328 and 12,348 patients were allocated to the development and validation cohorts, respectively.

The demographic characteristics of patients in both cohorts are summarised in Table 1. Emergency thoracotomy or laparotomy was performed in 582 (4.4%) patients in the development cohort and in 484 (3.9%) in the validation cohort. While comparable number of patients underwent emergency hemostatic surgeries in both cohorts (1.1% vs 1.0%), the validation cohort had slightly more patients (4.1%) who had emergency hemostatic IVR compared to the development cohort (3.3%). The validation cohort also had more patients (94.6%) who survived to be discharged to either their homes or other healthcare facilities compared to the development cohort (91.9%); hospital-free days till day 90 were similar between cohorts. Of patients in both cohorts, about 30% suffered from motor vehicle collisions and approximately 15% required transfusion within 24 hours after hospital arrival. The median ISS was 14 in both cohorts.

### Score development

Using the development cohort, multivariate logistic regression models were constructed with nine potential predictors of emergency surgery in patients who suffered from MVCs. The potential predictors were selected only from variables that could be obtained without detailed examination such as radiographic imaging, considering the clinical applicability. Prehospital vital signs were not selected as potential predictors due to considerable number of missing data (3301 [24.8%] cases), while vital signs upon arrival were selected. Cases with missing values (1298 [9.7%]) on selected potential predictors were replaced with estimated values using multiple imputations. Among the potential predictors examined, six independent predictors, namely, age, MOI (motor vehicle collision) and vital signs upon hospital arrival including GCS, RR, HR and sBP, were identified as predictor variables (Table 2). As interactions between age and GCS, as well as sBP and GCS, RR and HR, were expected, the interaction terms for these factors were analysed in regression models; however, there was no significant interaction between the predictors as no interaction terms with $p < 0.01$ were detected. Further, no

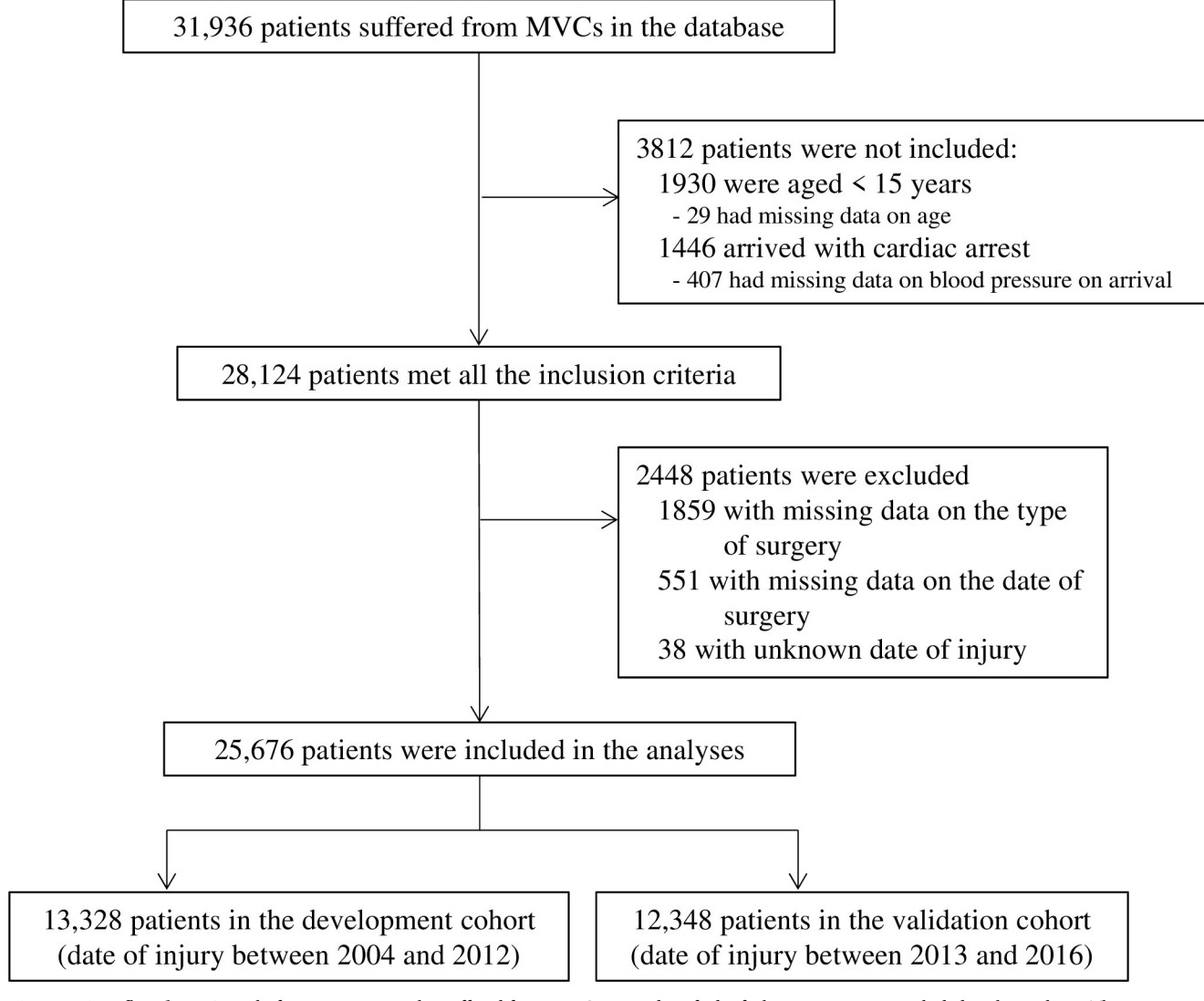

**Fig 1. Patient flowchart.** A total of 31,986 patients who suffered from MVCs were identified, of whom 25,676 were included in the analyses. The development cohort comprised 13,328 patients (those who were injured between 2004 and 2012) and the validation cohort comprised 12,348 patients (those who were injured between 2013 and 2016). Abbreviations: MVC = motor vehicle crashes.

significant multicollinearity was identified among the six identified predictors. Significant correlations between each prehospital vital sign (sBP, RR, or HR) and one upon hospital arrival were detected (S1 Table).

Based on the ORs of six predictors, scores in the SIM score were derived using weighted averages. Until the scores of the final model were obtained after the second adjustment of ORs, the c-statistic remained unchanged (0.796 in unadjusted model *vs*. 0.795 in both first adjusted and final models). Accordingly, the final model of the grading index, SIM score, was developed as 0 to 9 scales (Table 3).

### Score validation

ROC curves for SIM scores using the validation cohort revealed enough discrimination power (c-statistic = 0.789; 95% confidence interval [CI] = 0.768–0.810; $p < 0.01$), and multivariate

**Table 1. Characteristics of patients in development and validation cohorts.**

| | | Development Cohort | | Validation Cohort | |
|---|---|---|---|---|---|
| *n* | | 13,328 | | 12,348 | |
| Emergency thoracotomy or laparotomy, *n(%)* | | 582 | (4.4%) | 484 | (3.9%) |
| | Emergency thoracotomy, *n(%)* | 135 | (1.0%) | 80 | (0.6%) |
| | Emergency laparotomy, *n(%)* | 486 | (3.6%) | 427 | (3.5%) |
| Emergency hemostatic surgery, *n(%)* | | 143 | (1.1%) | 122 | (1.0%) |
| Emergency hemostatic IVR, *n(%)* | | 439 | (3.3%) | 506 | (4.1%) |
| | Additional hemostatic IVR, *n(%)* | 31 | (0.2%) | 40 | (0.3%) |
| Survival to discharge, *n(%)* | | 11,435 | (91.9%) | 11,414 | (94.6%) |
| missing data | | 884 | (6.6%) | 283 | (2.3%) |
| Hospital-free days to day 90, days, *median(IQR)* | | 13 | (29) | 13 | (26) |
| missing data | | 906 | (6.8%) | 289 | (2.3%) |
| Age, years, *median(IQR)* | | 47 | (38) | 53 | (39) |
| missing data | | 0 | (0.0%) | 0 | (0.0%) |
| Sex, male, *n(%)* | | 9323 | (70.0%) | 8275 | (67.0%) |
| missing data | | 1 | (0.0%) | 3 | (0.0%) |
| Mechanism of injury, *n(%)* | | | | | |
| | Motor vehicle collision | 3795 | (28.5%) | 3957 | (32.0%) |
| | Motorcycle collision | 4522 | (33.9%) | 3602 | (29.2%) |
| | Pedestrian-auto collision | 2394 | (18.0%) | 2269 | (18.4%) |
| | Bicycle collision | 2509 | (18.8%) | 2406 | (19.5%) |
| | Others | 108 | (0.8%) | 114 | (0.9%) |
| Vital signs upon hospital arrival | | | | | |
| | GCS, *median(IQR)* | 15 | (2) | 15 | (1) |
| | missing data | 413 | (3.1%) | 527 | (4.3%) |
| | Respiratory rate, /min, *median(IQR)* | 20 | (7) | 20 | (6) |
| | missing data | 980 | (7.4%) | 1024 | (8.3%) |
| | Heart rate, /min, *median(IQR)* | 84 | (25) | 82 | (23) |
| | missing data | 168 | (1.3%) | 97 | (0.8%) |
| | Systolic blood pressure, mmHg, *median(IQR)* | 132 | (38) | 136 | (37) |
| | missing data | 0 | (0.0%) | 0 | (0.0%) |
| Transfusion within 24 hours after arrival, *n(%)* | | 2087 | (15.9%) | 1843 | (15.1%) |
| missing data | | 222 | (1.7%) | 108 | (0.9%) |
| ISS, *median(IQR)* | | 14 | (15) | 14 | (13) |
| missing data | | 96 | (0.7%) | 183 | (1.5%) |

IQR = Interquartile Range, GCS = Glasgow Coma Scale, ISS = Injury Severity Score, IVR = Interventional Radiology

logistic regression also indicated good calibration (Hosmer–Lemeshow goodness-of-fit p = 0.124).

The SIM score-based estimate of the requirement of emergency surgery and the corresponding observed value for each point in the score are provided in Fig 2. Both estimated and observed probability of the need for emergent thoracotomy and/or laparotomy gradually increased from 1% at a score ≤ 1 to approximately 10% at a score of 5 and to almost 40% at a score ≥ 8.

To confirm that SIM score was independently associated with the need for emergent surgery, sensitivity analysis using multivariate logistic regression with SIM score and other clinical variables was performed, and the results show that validation was not dependent on the

**Table 2. Multivariate analysis for potential predictors of emergency surgical intervention.**

| | | Odds Ratio | 95% CI | P value |
|---|---|---|---|---|
| Age, ≥80 years | | 1.46 | 1.06–2.01 | 0.02 |
| Sex, male | | 1.17 | 0.96–1.43 | 0.12 |
| Mode of injury | | | | |
| | Motor vehicle collision | 6.28 | 4.31–9.14 | <0.01 |
| | Motorcycle collision | 2.81 | 1.90–4.15 | <0.01 |
| | Pedestrian-auto collision | 2.20 | 1.45–3.33 | <0.01 |
| Vital signs upon hospital arrival | | | | |
| | GCS, ≤13 | 1.57 | 1.30–1.91 | <0.01 |
| | RR, ≥22 /min | 1.66 | 1.39–2.00 | <0.01 |
| | HR, ≥100 /min | 2.00 | 1.66–2.41 | <0.01 |
| | sBP, <100 mmHg | 4.97 | 4.16–5.94 | <0.01 |

CI = Confidence interval, GCS = Glasgow Coma Scale, RR = respiratory rate, HR = heart rate, sBP = systolic blood pressure

method of comparison between the expected and the observed mortality (OR = 1.40 for 1 point increase in SIM score; 95% CI = 1.33–1.48; p < 0.01; Table 4).

Analyses on secondary outcomes identified that requirement of emergency hemostatic surgery was also significantly associated with SIM scores (OR = 1.19; 95% CI = 1.08–1.31; p <0.01; Table 4), as well as survival to discharge (OR = 0.81; 95% CI = 0.77–0.86; p < 0.01; Table 4). Requirement of emergency hemostatic IVR and hospital-free days till day 90 were not associated with SIM scores.

## Discussion

Here, we report on the development and validation of a novel and specific grading scale, the SIM score, for patients with MVC-related injury using independent cohorts. Notably, the SIM score could accurately predict the requirement for emergency surgery for victims of MVCs as a stepwise progression (linear calibration plots shown in Fig 2) with sufficient discriminatory power. Furthermore, sensitivity analyses and secondary outcome analyses revealed the relationship between SIM scores and the need for surgical intervention, as well as the requirement of emergency hemostatic surgery, indicating that the validation processes of SIM score were not dependent on the statistical approach used for comparison. Although an appropriate cut-off value for the SIM score would differ depending on the trauma system, it should be noted that less than 3 of the SIM score suggests < 5% of possibility of the need for emergency surgical intervention.

**Table 3. SIM score.**

| | Score |
|---|---|
| Age ≥80 years | 1 |
| Motor vehicle collision | 2 |
| GCS ≤13 on arrival | 1 |
| RR ≥22 /min on arrival | 1 |
| HR ≥100 /min on arrival | 1 |
| sBP <100 mmHg on arrival | 3 |

GCS = Glasgow Coma Scale, RR = respiratory rate, HR = heart rate, sBP = systolic blood pressure

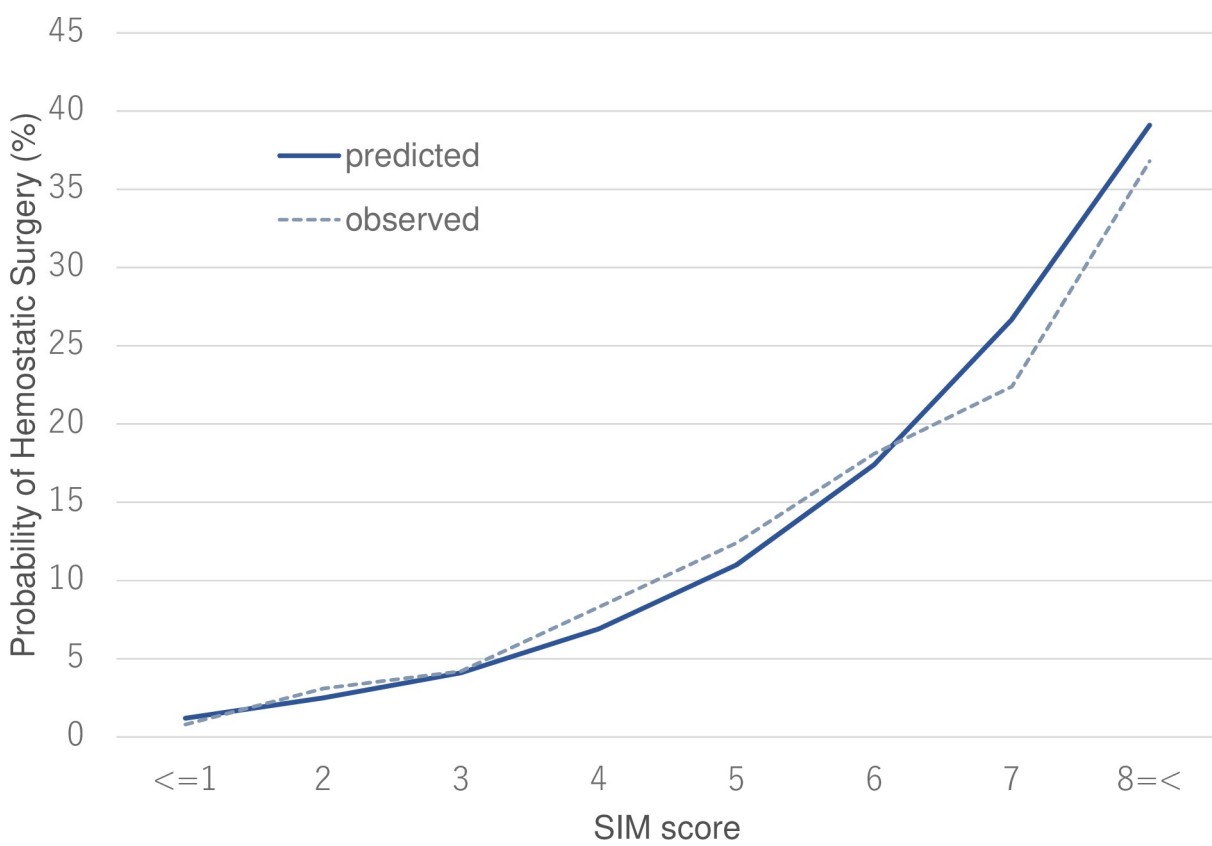

**Fig 2. SIM score-estimated and observed probabilities of the requirement for emergency surgical intervention.** Both estimated and observed probabilities of the requirement for emergency surgical intervention including thoracotomy and/or laparotomy gradually increased from 1% at a SIM score ≤ 1 to approximately 10% at a SIM score of 5 and to almost 40% at a SIM score ≥ 8. The linear calibration plots at each point of the SIM score reveal the plausibility of the index.

While well-developed trauma systems along with designated trauma centres have been validated to decrease mortality of severely injured patients [5,6,28], several challenges, such as fragmented distribution of surgeons and inappropriate patient allocation, were reported from regions with relatively immature trauma systems [29–31]. Although some scoring systems for

**Table 4. Impact of SIM score in sensitivity and secondary outcome analyses.**

|  | Odds Ratio[a] | 95% CI | P value |
|---|---|---|---|
| Emergency surgery | 1.40 | 1.33–1.48 | <0.01 |
| Emergency hemostatic surgery | 1.19 | 1.08–1.31 | <0.01 |
| Emergency hemostatic IVR | 0.98 | 0.93–1.03 | 0.41 |
| Survival to discharge | 0.81 | 0.77–0.86 | <0.01 |
|  | Coefficients |  |  |
| Hospital-free days to day 90, days[b] | 0.41 | −0.47–1.29 | 0.36 |

Multivariate logistic regression and multiple linear regression were used with adjustment for ISS and transfusion requirement within 24 hours after hospital arrival. CI = Confidence Interval, IVR = Interventional Radiology

[a]Odds Ratio for 1 point increase in SIM score

[b]days increase for 1 point increase in SIM score

severity of trauma patients, such as Revised Trauma Score and Probability of Survival determined by Trauma and Injury Severity Score, have been examined to develop a better triage algorithm that predicts emergency surgical intervention [18,19,32], the lower discrimination power at lower level trauma centres was identified and some of these systems failed to predict the requirement for emergency procedures including surgery [32–34]. Considering that the SIM score might independently predict the need for emergency thoracotomy and/or laparotomy regardless of injury severity (OR 1.40 for 1-point increase in SIM score; 95% CI = 1.33–1.48; Table 4), the SIM score can be implemented even in non-designated trauma centres to where patients with less severe injuries are transported, and would help deciding to transfer patients to other hospitals with surgical capabilities.

During the development phase of this study, we analysed the variables used in the existing severity scales or potential prognostic indicators that had been independently assessed in trauma patients [19–23]. Considering that this process satisfied pathophysiological plausibility and was predicated by previous studies, we believe that our novel scoring index can be adopted across trauma centres in different countries for the benefit of MVC victims and patients. It should also be noted that the SIM score was constructed utilising simple vital signs along with age and motor vehicle collisions as MOI, and therefore, would be a user-friendly scoring system.

The results of this study must be interpreted in the context of the study design. We only included patients who suffered from MVCs, which may limit the generalizability of our findings. However, as MVC is main cause of blunt injury across the world and penetrating injury has been validated to be a robust predictor for emergency procedures [1,2,4,34,35], we believe that the SIM score would be practical and applicable in most regions.

Another limitation is the fact that prehospital vital signs were not incorporated into the initial regression model as potential predictors due to high number of missing data. As the vital sings upon hospital arrival are not always similar to prehospital measurements made at the scene of injury, this might gravely affect the validity of the SIM score to use as a prehospital triage algorithm. Although the significant correlations between prehospital vital signs and those upon hospital arrival were identified, the SIM score would only holds promise for the effective determination of early transfer or activation of surgical subspecialties after hospital arrival until the score is externally validated with other data sets in future studies.

Furthermore, we defined several variables as emergency procedures when they were performed within 24 hours after hospital arrival. As patients who needed surgical intervention several hours after arrival might not have had fatal haemorrhage, the SIM score would not exclusively predict immediate hemostatic intervention for exsanguinating injury. Whether the SIM score would predict the need for resuscitative hemostatic procedure, as well as whether applying the score would introduce favorable clinical outcome, should be further examined.

Finally, because this is a retrospective study, the results are not conclusive. Residual cofounding and unmeasured predictors for surgical intervention, such as fluid resuscitation and responsiveness, duration from injury to hospital arrival and medications administered prior to hospital arrival, are impediments that prevent confirming an association between SIM score and the requirement for emergency surgical interventions. Although we have validated the SIM score with multiple analyses in the validation cohort, additional clinical investigations, including a prospective observational study, must be conducted to validate our findings.

In conclusion, SIM score was developed and validated as a novel scoring system for predicting the requirement for emergency surgical interventions. The SIM score is a user-friendly grading index that could effectively estimate the need for emergency thoracotomy and/or laparotomy as a stepwise progression.

## Supporting information

**S1 Table. Correlations between vital signs at prehospital and upon hospital arrival.**
(DOCX)

## Author Contributions

**Conceptualization:** Ryo Yamamoto.

**Data curation:** Ryo Yamamoto.

**Formal analysis:** Ryo Yamamoto, Junichi Sasaki.

**Investigation:** Ryo Yamamoto, Junichi Sasaki.

**Methodology:** Ryo Yamamoto, Tomohiro Kurihara, Junichi Sasaki.

**Project administration:** Tomohiro Kurihara.

**Supervision:** Tomohiro Kurihara, Junichi Sasaki.

**Validation:** Junichi Sasaki.

**Writing – original draft:** Ryo Yamamoto.

**Writing – review & editing:** Ryo Yamamoto, Tomohiro Kurihara, Junichi Sasaki.

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
