## [Decision Letter · Decision Letter 0]

14 Nov 2019

PONE-D-19-24717

A novel scoring system to predict the requirement for surgical intervention in victims of motor vehicle crashes: Development and validation using independent cohorts

PLOS ONE

Dear Dr. YAMAMOTO,

Thank you for submitting your manuscript to PLOS ONE. After careful consideration, we feel that it has merit but does not fully meet PLOS ONE’s publication criteria as it currently stands. Therefore, we invite you to submit a revised version of the manuscript that addresses the points raised during the review process.

We would appreciate receiving your revised manuscript by Dec 29 2019 11:59PM. To enhance the reproducibility of your results, we recommend that if applicable you deposit your laboratory protocols in protocols.io, where a protocol can be assigned its own identifier (DOI) such that it can be cited independently in the future. For instructions see: http://journals.plos.org/plosone/s/submission-guidelines#loc-laboratory-protocols

We look forward to receiving your revised manuscript.

Kind regards,

Itamar Ashkenazi

Academic Editor

PLOS ONE

Journal Requirements:

2. Please note that PLOS journals require authors to make all data underlying the findings described in their manuscript fully available without restriction at the time of publication. When specific legal or ethical requirements prohibit public sharing of a dataset, authors must indicate how researchers may obtain access to the data. PLOS journals will not consider manuscripts for which the following factors influence ability to share data:

- Authors will not share data because of personal interests, such as patents or potential future publications.

- The conclusions depend solely on the analysis of proprietary data, whether these data are owned by the authors, by their funders or institutions, or by other parties.

Therefore, please update your Data Availability statement to indicate how other researchers may gain access to the underlying data reported in the manuscript. For more information, please see: https://journals.plos.org/plosone/s/data-availability

3. Thank you for including your ethics statement:

"All collaborating hospitals obtained individual local institutional review board approval for the Conduct of Human Research before initiating the study (approval number: 20090087 at Keio University School of Medicine).".

Additional Editor Comments (if provided):

Dear Author,

Three very experienced and distinguished trauma physicians reviewed your manuscript. Their recommendations on whether to accept or not to accept this manuscript for publication were far from being homogenous. Attached to this letter below, you will find their comments that need to be addressed. Among the criticism raised, there is one major limitation I wish to discuss. The score significantly relies on data that was measured upon admission of the patients to the emergency department (3 of 6 variables accounting for 6 of 9 potential points). You were forced to make this decision since prehospital vital-signs were missing in many of your study subjects. This gravely affects the validity of your results. One cannot assume that the measurements upon admission to the hospital are similar to prehospital measurements made at the scene of injury. Since the score aims to guide the prehospital team as to where to transport the trauma victim, this deficiency is of utmost importance. Still, it does not rule out this study. This deficit, however, should be discussed thoroughly as a limitation at the appropriate place within the manuscript.

Sincerely,

Itamar Ashkenazi, Academic Editor

Reviewers' comments:

Reviewer's Responses to Questions

**Comments to the Author**

1. Is the manuscript technically sound, and do the data support the conclusions?

Reviewer #1: No

Reviewer #2: Yes

Reviewer #3: Yes

2. Has the statistical analysis been performed appropriately and rigorously? 

Reviewer #1: Yes

Reviewer #2: Yes

Reviewer #3: I Don't Know

3. Have the authors made all data underlying the findings in their manuscript fully available?

Reviewer #1: Yes

Reviewer #2: Yes

Reviewer #3: Yes

4. Is the manuscript presented in an intelligible fashion and written in standard English?

Reviewer #1: Yes

Reviewer #2: Yes

Reviewer #3: Yes

5. Review Comments to the Author

Reviewer #1: the authors have attempted to create a prehospital score that will help predict who needs "emergency surgery" using data from a registry. I understand the need, and agree, however I have several significant concerns.

1. emergency surgery is broader than thoracotomy or laparotomy. For instance why isnt major vascular repair or craineotomy considered emergency surgery?

2. the use of 24 hours to define multiple time points in the paper is a major problem. It likely springs from using a registry, and times within 60 minutes of arrival may not be available. emergency surgery performed 18 hours after injury is likely not important to survival. These times after injury have been explored by multiple authors.

3. the lack of prehospital vital signs in a prehospital score is really an issue.

Reviewer #2: Thanks for submitting you score derivation study. The methodology and sample size calculation are noted. The grammar is generally good - there are minor English errors that do not really detract from the message. The statistical analysis seems in order and the message is clear - the predicted and actual rates were clearly similar.

This is a good study and results in a practical score that should now be externally validated.

Reviewer #3: This is an interesting study which probably has some local relevance. The broader issue is whether this score will be helpful. Clearly at present trauma patients are taken to hospitals without surgical capabilities. This is hardly every appropriate and I presume this happens because the prehospital resources do not have the capability to take trauma patients to hospitals with surgical capabilities. In that context, will knowing that they are more likely to need surgery be helpful?

Some specific comments are

Language such as "trauma care should imbibe the philosophy of hemostasis" might be appropriate in a novel but is probably too flowery for a scientific paper

The outcome measure used (which is a good one) is usually described as "days alive and out of hospital (DAOH)

Why was 80yo chosen as the cut score for age

Is it helpful to use vital signs on arrival when clearly it would be better if the patients did not arrive at hospitals without surgical capability. If poor or missing data is an issue in relation to prehospital vital signs would that not be an important issue to address?

It looks like the likelihood of needing surgery is 5% with a score of 3, 7.5% with 4, 15% with 5, 20% with 6, 25% with 7 and 40% with 8. What is the cut point which the authors feel justifies transport to a surgical facility?

6. PLOS authors have the option to publish the peer review history of their article (what does this mean?). If published, this will include your full peer review and any attached files.

Reviewer #1: No

Reviewer #2: No

Reviewer #3: No

---

## [Author Response · Author response to Decision Letter 0]

19 Nov 2019

Editor

Author Response: Thank you and your reviewers for the insightful comments and the opportunity to resubmit our manuscript. We have revised the manuscript following your comments. Please see our specific responses below.

1. The score significantly relies on data that was measured upon admission of the patients to the emergency department (3 of 6 variables accounting for 6 of 9 potential points). You were forced to make this decision since prehospital vital-signs were missing in many of your study subjects. This gravely affects the validity of your results. One cannot assume that the measurements upon admission to the hospital are similar to prehospital measurements made at the scene of injury. Since the score aims to guide the prehospital team as to where to transport the trauma victim, this deficiency is of utmost importance. Still, it does not rule out this study. This deficit, however, should be discussed thoroughly as a limitation at the appropriate place within the manuscript.

Author Response: Thank you for bringing this to our attention. We agree with your comments that not including prehospital vital signs would be a significant limitation. To clarify this limitation, the text in lines 334-340 in p.21 in Discussion have been revised to the following:

“Another limitation is the fact that prehospital vital signs were not incorporated into the initial regression model as potential predictors due to high number of missing data. As the vital sings upon hospital arrival are not always similar to prehospital measurements made at the scene of injury, this might gravely affect the validity of the SIM score to use as a prehospital triage algorithm. Although the significant correlations between prehospital vital signs and those upon hospital arrival were identified, the SIM score would only holds promise for the effective determination of early transfer or activation of surgical subspecialties after hospital arrival until the score is externally validated with other data sets in future studies.”

 

Reviewer 1

Author Response: Thank you for your attention to detail and requests for clarification. We are grateful for your insightful comments and the valuable suggestions on our manuscript. We have revised the manuscript following your comments. Please see our specific responses below. 

1. Emergency surgery is broader than thoracotomy or laparotomy. For instance why isn’t major vascular repair or craniotomy considered emergency surgery?

Author Response: Thank you for pointing this out to us. Our writing was unclear. We didn’t consider craniotomy and vascular repair as emergency surgical intervention because some anatomic and physiologic criteria, such as Glasgow Coma Scale (GCS) and massive bleeding and/or distal circulatory deficit in extremities have been reported to be able to predict the need for such surgical interventions in previous studies. We added the references and the text have been revised to the following:

In line 66 in p.4 in Introduction, “Although several triage algorithms using mechanism of injury (MOI) and/or anatomic and physiologic criteria successfully identified patients who needed treatment at high-level trauma centres [11-13], they have not been validated to predict the need for emergency surgical intervention, particularly for laparotomy and thoracotomy.”

In lines 106-110 in p.6 in Methods, “Craniotomy and vascular surgery in extremities were not included as the emergency surgical intervention in this study because some anatomic and physiologic criteria, such as Glasgow Coma Scale (GCS) and massive bleeding and/or distal circulatory deficit in extremities have been reported to be able to predict the need for such surgical interventions [16,17].”

2. The use of 24 hours to define multiple time points in the paper is a major problem. It likely springs from using a registry, and times within 60 minutes of arrival may not be available. emergency surgery performed 18 hours after injury is likely not important to survival. These times after injury have been explored by multiple authors

Author Response: Thank you for this important feedback. We agree with your comments and understand that the SIM score would not exclusively predict immediate hemostatic intervention for exsanguinating injury and would not be associated with clinical outcomes. To clarify this limitation, we added a paragraph in lines 341-347 in pp.21-22 in Discussion as follows:

“Furthermore, we defined several variables as emergency procedures when they were performed within 24 hours after hospital arrival. As patients who needed surgical intervention several hours after arrival might not have had fatal haemorrhage, the SIM score would not exclusively predict immediate hemostatic intervention for exsanguinating injury. Whether the SIM score would predict the need for resuscitative hemostatic procedure, as well as whether applying the score would introduce favorable clinical outcome, should be further examined.”

3. The lack of prehospital vital signs in a prehospital score is really an issue.

Author Response: Thank you for bringing this to our attention. We agree with your comments that not including prehospital vital signs would be a significant limitation. To clarify this limitation, the text in lines 334-340 in p.21 in Discussion have been revised to the following:

“Another limitation is the fact that prehospital vital signs were not incorporated into the initial regression model as potential predictors due to high number of missing data. As the vital sings upon hospital arrival are not always similar to prehospital measurements made at the scene of injury, this might gravely affect the validity of the SIM score to use as a prehospital triage algorithm. Although the significant correlations between prehospital vital signs and those upon hospital arrival were identified, the SIM score would only holds promise for the effective determination of early transfer or activation of surgical subspecialties after hospital arrival until the score is externally validated with other data sets in future studies.”

 

Reviewer 2

Author Response: We are grateful for your comments on our manuscript. We will validate the SIM score externally using other dataset in a future study.

 

Reviewer 3

Author Response: Thank you for your valuable comments on our manuscript and requests for clarification. We have revised the manuscript following your comments. Please see our specific responses below. 

1. The broader issue is whether this score will be helpful. Clearly at present trauma patients are taken to hospitals without surgical capabilities. This is hardly every appropriate and I presume this happens because the prehospital resources do not have the capability to take trauma patients to hospitals with surgical capabilities. In that context, will knowing that they are more likely to need surgery be helpful?

Author Response: Thank you for bringing this to our attention. We think the score needs to be externally validated to use as a prehospital triage algorithm, particularly outside the study setting. To clarify that the current study would only conclude the SIM score would help deciding to transfer patients to other hospitals with surgical capabilities, the text in Discussion have been revised to the following:

In lines 316-317 in p.20, “the SIM score can be implemented even in non-designated trauma centres to where patients with less severe injuries are transported, and would help deciding to transfer patients to other hospitals with surgical capabilities.”

In lines 337-340 in p.21, “the SIM score would only holds promise for the effective determination of early transfer or activation of surgical subspecialties after hospital arrival until the score is externally validated with other data sets in future studies.”

2. Language such as "trauma care should imbibe the philosophy of hemostasis" might be appropriate in a novel but is probably too flowery for a scientific paper.

Author Response: Thank you, this has been corrected as follows.

In line 60 in p.4 in Introduction, “Considering that timely surgical interventions improve outcomes for trauma victims following MVCs, the strategy for trauma care should include early hemostasis, decontamination of peritoneal infection, and early control of digestive fluid leakage, particularly outside the well-designed trauma system or designated trauma centres.”

3. The outcome measure used (which is a good one) is usually described as "days alive and out of hospital (DAOH).

Author Response: Thank you for pointing this out to us. After discussing among authors, we decided to use “hospital-free days”. However, we added “days alive and out of hospital” to explain the hospital free days. The text was added in line 120 in p.7 in Methods as follows:

“The primary outcome was the requirement of emergency surgery. Secondary outcomes included emergency hemostatic surgery, emergency hemostatic IVR, survival to discharge, and hospital-free days, also known as days alive and out of hospital, till day 90 after injury (combination of in-hospital death and length of hospital stay defined as the number of days alive and out of the hospital).”

4. Why was 80yo chosen as the cut score for age?

Author Response: Thank you for the valuable comment. We used the cut-off to define “old-old” elderly population because some previous studies suggested old-old patients aged ≥80 years had different outcomes. We added a reference (#23) in line 145 in p.8 in Methods as follows:

“Owing to clinical plausibility and practicality, age and vital signs were transformed to binary variables as follows: age (< 80 or ≥80 years) [23], GCS(≤13 or ≥14), systolic blood pressure (sBP; <110 or ≥110 mmHg) [22], heart rate (HR; <100 or ≥100 bpm), respiratory rate (RR; <22 or ≥22 /min).”

5. Is it helpful to use vital signs on arrival when clearly it would be better if the patients did not arrive at hospitals without surgical capability. If poor or missing data is an issue in relation to prehospital vital signs would that not be an important issue to address?

Author Response: Thank you for bringing this to our attention. We agree with your comments that not including prehospital vital signs would be a significant limitation. To clarify this limitation, the text in lines 334-340 in p.21 in Discussion have been revised to the following:

“Another limitation is the fact that prehospital vital signs were not incorporated into the initial regression model as potential predictors due to high number of missing data. As the vital sings upon hospital arrival are not always similar to prehospital measurements made at the scene of injury, this might gravely affect the validity of the SIM score to use as a prehospital triage algorithm. Although the significant correlations between prehospital vital signs and those upon hospital arrival were identified, the SIM score would only holds promise for the effective determination of early transfer or activation of surgical subspecialties after hospital arrival until the score is externally validated with other data sets in future studies.”

6. It looks like the likelihood of needing surgery is 5% with a score of 3, 7.5% with 4, 15% with 5, 20% with 6, 25% with 7 and 40% with 8. What is the cut point which the authors feel justifies transport to a surgical facility?

Author Response: Thank you for your thoughts about this important component of our manuscript. Although an appropriate cut-off value for the SIM score would differ depending on the trauma system, we believe 3 would be low enough cut-off value because it suggests < 5% of possibility of the need for emergency surgical intervention. We added sentences in lines 300-302 in p.19 in Discussion as follows:

“Although an appropriate cut-off value for the SIM score would differ depending on the trauma system, it should be noted that less than 3 of the SIM score suggests < 5% of possibility of the need for emergency surgical intervention.”

---

## [Editor Report · Decision Letter 1]

25 Nov 2019

A novel scoring system to predict the requirement for surgical intervention in victims of motor vehicle crashes: Development and validation using independent cohorts

PONE-D-19-24717R1

Dear Dr. YAMAMOTO,

Thank you for addressing the reviewers' comments.

We are pleased to inform you that your manuscript has been judged scientifically suitable for publication and will be formally accepted for publication once it complies with all outstanding technical requirements.

With kind regards,

Itamar Ashkenazi

Academic Editor

PLOS ONE
---

## [Editor Report · Acceptance letter]

2 Dec 2019

PONE-D-19-24717R1 

A novel scoring system to predict the requirement for surgical intervention in victims of motor vehicle crashes: Development and validation using independent cohorts 

Dear Dr. YAMAMOTO:

I am pleased to inform you that your manuscript has been deemed suitable for publication in PLOS ONE. Congratulations! Your manuscript is now with our production department. 

With kind regards,

on behalf of

Dr. Itamar Ashkenazi 

Academic Editor

PLOS ONE